# Evaluating the content and face validity of Arabic-translated Patient Measures of Safety survey PMOS-30

**Yasser A. Alaska[1,2], Nawaf M. Alqahtani[2], Amani K. Al Zahrani[2], Rawan Alshahri[2], Rana Z. Malyani[3], Rabab B. Alkutbe[2]***

**1** Technical Affair, Saudi Patient Safety Center (SPSC), Riyadh, Saudi Arabia, **2** Department of Emergency Medicine, College of Medicine, King Saud University, Riyadh, Saudi Arabia, **3** Medical College, Umm Al-Qura University, Makkah, Saudi Arabia

\* dr.rabab.alkutbe@hotmail.com, ralkutbe@spsc.eu.sa

## Abstract

### Background

The importance of patient perception of patient safety has been proven as an active role in promoting safety and predicting harm. Patients play a vital role in the healthcare sector and the impact of patient perception of patient safety has been repetitively proven in the literature to be for its effectiveness in predicting harm and promoting safety. Nonetheless, there is limited knowledge about the specific insights patients can provide concerning safety within Arab countries. Therefore, improving and strengthening active patient participation by including patients' perceptions of safety could offer novel contributions to the realm of patient safety.

### Objective

This study aims to evaluate the validity of the Arabic version of the PMOS-30.

### Method

The forward-backward translation procedure was used to translate and validate the PMOS-30. Mix methods were used to assess the validity of the translated version. The expert raters evaluated the content validity and interviews were conducted with in-patients to assess the face validity. Test-retest approach was conducted to pilot the final Arabic version.

### Results

Data of face validity from 13 participants for the first test and 5 participants for the re-test was collected and showed an improvement in the clarity rate (71.50% and 94.66% respectively). With respect to the content validity of the final version, the CVI was 0.92, indicating excellent relevant results.

**Data Availability Statement:** All relevant data are within the manuscript.

**Funding:** The authors received no specific funding for this work.

**Competing interests:** The authors have declared that no competing interests exist.

## Conclusion

The final version of the revised was approved by the expert to be a valid tool to measure patient perceptions of patient safety in Arabic-speaking patients to be utilized on patient safety improvement initiatives.

## Introduction

Patient safety is a worldwide concern, and according to the World Health Organization (WHO), many patients received unsafe medical care, which resulted in disabling injuries or death [1]. The challengeable initiatives were developed to measure and improve patient safety levels globally [2]. It has been found that patient engagement is an essential element that can enhance safety aspects and improve the healthcare system [3].

The recognition of the difficulties associated with implementing patient engagement to measure patient safety is investigated; however, the significance of patient involvement lies in capturing measures that signify safety improvement and harm reduction [4]. Although there is limited data on patient engagement in medical care in the Arab world, the need to measure patient safety through patient perception will enhance patient empowerment for various stakeholders. Despite the fact that the patient's involvement plays a crucial role in safety monitoring and improvement, the adaptation of their resources is underutilized [5]. Accordingly, numerous instruments were introduced to measure safety from a patient's perspective and were widely used in various research endeavors to encompass all aspects of patient safety for the enhancement of healthcare quality [6].

Patient satisfaction feedback on their experiences in receiving care is now commonly used in the UK; thus, many significant reports have emphasized the importance of considering patients' concerns to improve the quality of care [7, 8].

Many interventions are conducted based on theories and evidence-based approaches that lay on collecting hospital in-patient feedback regarding safety. This process helps to develop a proactive assessment tool that systematically measures safety from the patient perspective.

Several survey tools were developed to measure patient satisfaction or perceptions of safety culture and have been widely used. However, there is a limited Arabic tool that helps improve patient safety from the patients' feedback on the safety of their care environment.

Finding tools to assess patient safety from the perspective of Arabic-speaking patients posed a challenge, as most of the available tools focused on measuring patient satisfaction or patient experience during the care visit. However, at the international level, tools have been developed in the English language specifically to assess patient safety. One such tool is the Primary Care Patient Safety Scale (PC PMOS), which aims to gather patient feedback on factors contributing to patient safety in primary care settings. The PC PMOS tool collects information regarding patients' perceptions of safety and helps identifying areas for improvement [9].

Another noteworthy tool is the PROSPER Consortium, which aims to enhance safety reporting by incorporating patient-reported outcome measures and perspectives. This initiative seeks to increase patient participation in identifying and reporting safety issues [10].

In addition, it is worth noting the inclusion of the Outpatient Adverse Event Trigger Tool, which aids in the identification of adverse events specifically in outpatient settings. Furthermore, the Agency for Healthcare Research and Quality (AHRQ) has developed various tools, including CAHPS (Consumer Assessment of Healthcare Providers and Systems), to assess

patient satisfaction. These surveys specifically target patients' experiences with healthcare providers, hospitals, and health plans [11].

Moreover, PROMIS, which stands for the Patient-Reported Outcomes Measurement Information System, is a person-centered set of measures developed by the National Institutes of Health (NIH) [12]. These measures are designed to evaluate and monitor physical, mental, and social health in adults and children. PROMIS assesses various aspects of health, including pain, fatigue, physical functioning, emotional distress, and social role participation with greater precision than conventional measures. This precision enhances research and clinical settings. Furthermore, PROMIS has been translated into different languages [13].

It is crucial to emphasize the initiatives and programs implemented by the Saudi Arabian Ministry of Health (MOH) to enhance the patient's experience. One notable program is the Patient Experience Measurement Program, which evaluates patient satisfaction across various levels of healthcare services. The program collects feedback from patients to understand their views on care quality, communication, access, and overall satisfaction. However, there is limited availability of Arabic tools that aim to assess patient safety from a patient perspective in this context.

The patient measure of safety (PMOS) tool was developed based on the contributory factors from the Yorkshire Contributory Factors Framework (YCFF) and focused on the ability of patients in a hospital setting to identify factors that contribute to safety [9]. The PMOS provides a way of systematically evaluating the contributing factors that highlight the safety concerns from the patient's perspective to help healthcare organizations improve care [14, 15]. The 44 items of PMOS were shortened and produced two versions of PMO (PMOS-30 and PMOS-10); accordingly, the validated version of PMOS-30s was used as a diagnostic function part of an intervention [16]. However, these tools are limited in languages; therefore, an Arabic version is needed to be used in Arab-speaking countries.

For instance, the translation procedure may pose challenges related to question phrasing and the length of the questions, which could potentially impact the reliability and validity of responses, potentially leading to missing data. To ensure the validity of the conceptual content of the translated tool, the items of the assessment tool must not only be translated linguistically but also culturally modified to be utilized adequately in other languages [17].

Therefore, the translated tool needs to be thoroughly checked and tested for linguistic and cultural differences to reach the acceptable level of reliability and validity of the content, semantic, technical, and conceptual translated tool [18].

Piloting the tool to ensure the validity of the results from the target sample is the preferred approach. Thus, an Australian study aimed to revise and pilot the PMOS tool to measure vulnerable and elderly patients' perception of safety in the hospital setting. The revised piloted tool was deployed as part of the Australian national study [19].

Testing the revised tool is a necessity in order to ensure the applicability and efficacy of the tool in obtaining the data from the target population. Therefore, this study aims to evaluate the reliability and validity of the translated version of the PMOS-30 to assess the patient perception of Patient Safety in Saudi Arabia.

## Methods

The study was conducted as a multi phases study, and methodological triangulation was used to combine methods in the validation phase [20]. The original PMOS 30 consists of 30 items based on 8 domains. Respondents address these 30 items by means of a five-point Likert scale of which the labels vary throughout the domains; 1 = 'strongly disagree' to 5 = 'strongly

agree. We have received permission to translate the original version of the PMOS-30 Questionnaire to the Arabic version from the National Health Service "NHS" UK.

## Translation procedure

The PMOS-30 Questionnaire was translated into Arabic using the common forward-backward translation method [21, 22].

An English draft version of PMOS-30 is provided to two native Arabic translators for forward translation. Then, both Arabic translators gave difficulty ratings during the forward translation process. The translators discussed standardizing the Arabic version. After the agreement on the final Arabic version, it was given to two native English translators for backward translation. At this step, the difficulty and quality ratings of the Arabic version are given by English translators during backward translation. Then, English translators finalized the final English version. The Quality ratings of the standard English version were given by two independent Arabic translators. At the final step, both final versions were reviewed by the Saudi Patient Safety Center team, Quality patient safety, and patient experience officers (Table 2).

## Validation procedure

**Content and face validity.** To evaluate the questions, each question was assessed to validate the content and the concept according to the framework from the NHS. Face and content validity were conducted to ensure the translated questions adequately measure the intended purpose by the expert panel using the guideline developed by Patel and Desai 2020 [23]. For our research, we selected 5 experts for the initial test and 3 experts for the retest. These experts were chosen from individuals who had previously participated in a similar panel, and all possessed postgraduate education, skills training, and an average of 9 years of patient experience in Arab countries. The composition of the panel members encompassed expertise in instrument development, healthcare provision, and health education.

The content validity was performed using raters' forms to assess the pre-final version of the questionnaire's validity and the percentage of agreement between them. To assess the validity of the responses, an overall agreement rating was conducted using a 4-point scale that considered factors such as relevance, clarity, simplicity, and ambiguity. The content validity index (CVI) was determined based on the judgment of the experts [23–25]. Then, the percentage of agreement and acceptability of each question was adjusted based on the classification of > 90% retain, 90–80% redefine, or < 80% restructure.

The qualitative approach was used to attain input from the service users and to assess the validity of the translated questionnaire among a sample of the targeted population. The number of participants for the interview was based on the recommendation of the sample size to achieve saturation of data [26]. The sufficient range is between 14 to 5 participants in the interview to achieve a mixed perspective and identify the issues in the tool.

A pilot test was conducted by interviewing the participants to evaluate each question in regard to capturing the topic under investigation and scribbling notes. During the interview, each participant pretended to fill out the survey and the trained researcher asked the participants whether any sentences were difficult to understand. The questionnaire also checked for any common errors like double-barreled, confusing, and length by using the three main criteria: clarity of the question, simplicity, and relevance. These three criteria are based on the dimension suggested by Flaherty et al. 1988 of assessing the content to determine if it is relevant, the semantic equivalence, which means each item is similar after translation and the conceptual equivalence [27, 28].

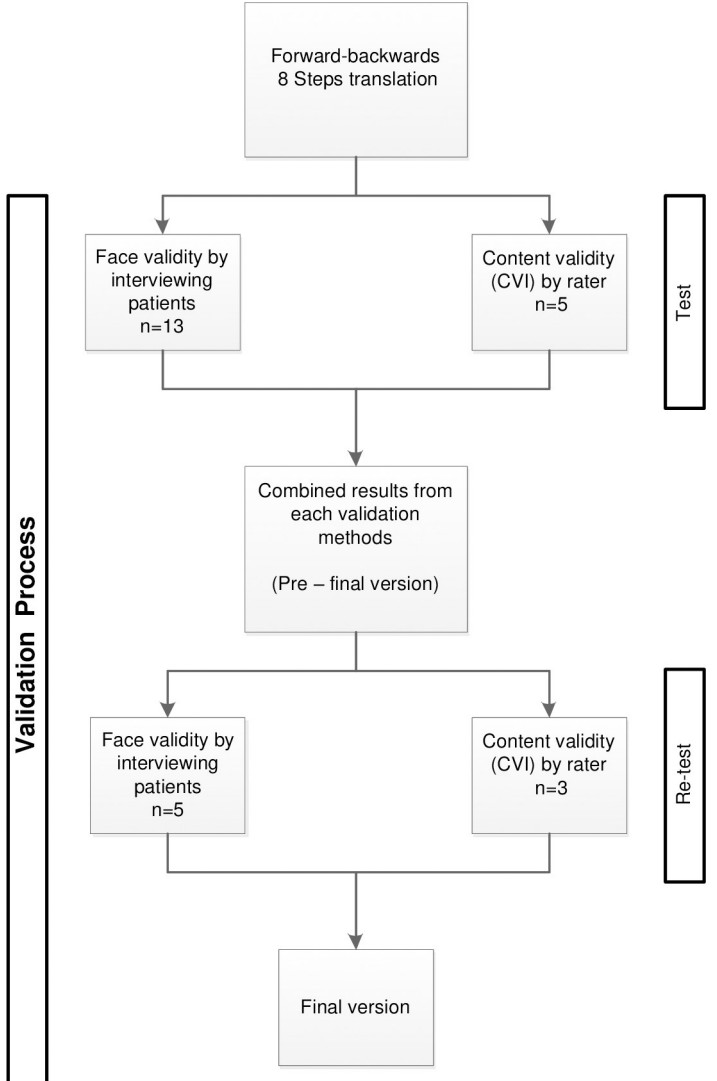

**Fig 1. Overview of the validation process of the PMOS-30 Arabic version.**

Questionable items or poor wordings were recorded and reported to the questionnaire translators for further discussion and revision. The last modifications based on the combination of both results were utilized, and the final version was retested for the second round of validation (Fig 1).

## Participants

Following the translation process, the modified questionnaire was piloted into two groups for testing and retesting for the second round of validation. The inclusion criteria required participants to be in-patients who had been discharged from the hospital within the past two weeks. In this pilot study, 13 patients (7 female and 6 male) aged between 18–70 years were included, and their consent was obtained. Trained researchers interviewed the participants either by telephone or via Zoom. In the re-test phase, 5 participants from the same population were interviewed and assessed the quality and comprehensibility of the questions in the final version of the translated tool.

## Statistical analysis

All statistical analyses were performed using R, an open-source programming environment, and Excel. To examine the content validity, the count of clarity agreement between the raters for each item was measured by a 4-point (CVI) [24]. The item content validity index (I-CVI) for each item was calculated as (count of clarity agreement/no. of raters) and with the average for the I-CVI.

Kappa ($k$) statistic was used to interpret the CVI values to capture and adjust the chance agreement for the inter-rater [29]. In order to estimate the Kappa, the probability of a chance occurrence ($P_c$) was first measured as

$$P_c = [N!/A!(N-A)!]*.5^n$$

The N is the number of raters, and A is the number agreeing on good relevance. Then Kappa was calculated as

$$k = (I-CVI-p_c)/(1-p_c)$$

and the considerations were as follows, $k < 0.40$ Poor, $> 0.40$ Fair, $> 0.6$ Good, and $> 0.74$ "Excellent [30, 31]. This R Language package utilized "tidyverse", Excel functions were used "IF","IFS","CountIF"," FACT", "Average"

## Ethical consideration

Ethical approval from King Saud University (E-21-6527) was obtained prior to the commencement of any study procedure. Participants were fully informed via a link about the study aims and the use of their data, and their consent was provided verbally before participating in the study.

## Results

The translation procedures are insufficient to detect the significant discrepancies in the Arabic Version; therefore, the pre-final Arabic version from the translation process was tested for face and content validity. The total number of participants who were interviewed in the first face validation test was 13 patients (Table 1).

The participants were inpatients from different socioeconomic statuses such as employment and educational level. Also, the participants varied in location, age, and services received.

**Table 1. Socio-demographic characteristics for test and re-test interview participant.**

| Socio-Demographic Characteristics | | Total responses test total = 13 n (%) | Total responses re-test total = 5 n (%) |
|---|---|---|---|
| Gender | Male | 6 (45%) | 3 (60%) |
| | Female | 7 (55%) | 2 (40%) |
| Age | 18–35 | 3 (23%) | 2 (40%) |
| | 36–55 | 9 (69%) | 3 (60%) |
| | 56–70 | 1 (8%) | 0 (0%) |
| Educational Level | Bachelor and above | 4 (30%) | 2 (40%) |
| | High School | 6 (46%) | 3 (60%) |
| | Intermediate School | 3 (23%) | 0 (0%) |
| Employment | Housewife | 5 (38%) | 2 (40%) |
| | Employed | 6 (45%) | 3 (60%) |
| | Unemployed | 2 (15%) | 0 (0%) |

**Table 2. Face validity and clarity rate of the questions based on the interview test and the re-test.**

| Question | Not Clear Count | | Clear Count | | Clarity Percentage* | |
|---|---|---|---|---|---|---|
| | Test[a] | Re-test[b] | Test | Re-test | Test | Re-test |
| Q1 | 1 | 0 | 12 | 5 | 92.31% | 100% |
| Q2 | 3 | 1 | 10 | 4 | 76.92% | 80% |
| Q3 | 2 | 0 | 11 | 5 | 84.62% | 100% |
| Q4 | 7 | 0 | 6 | 5 | 46.15% | 100% |
| Q5 | 10 | 0 | 3 | 5 | 23.08% | 100% |
| Q6 | 6 | 2 | 7 | 3 | 53.85% | 60% |
| Q7 | 6 | 0 | 7 | 5 | 53.85% | 100% |
| Q8 | 3 | 1 | 10 | 4 | 76.92% | 80% |
| Q9 | 7 | 0 | 6 | 5 | 46.15% | 100% |
| Q10 | 1 | 0 | 12 | 5 | 92.31% | 100% |
| Q11 | 3 | 0 | 10 | 5 | 76.92% | 100% |
| Q12 | 4 | 0 | 9 | 5 | 69.23% | 100% |
| Q13 | 6 | 0 | 7 | 5 | 53.85% | 100% |
| Q14 | 5 | 0 | 8 | 5 | 61.54% | 100% |
| Q15 | 2 | 0 | 11 | 5 | 84.62% | 100% |
| Q16 | 2 | 0 | 11 | 5 | 84.62% | 100% |
| Q17 | 2 | 1 | 11 | 4 | 84.62% | 80% |
| Q18 | 2 | 0 | 11 | 5 | 84.62% | 100% |
| Q19 | 1 | 0 | 12 | 5 | 92.31% | 100% |
| Q20 | 2 | 1 | 11 | 4 | 84.62% | 80% |
| Q21 | 3 | 0 | 10 | 5 | 76.92% | 100% |
| Q22 | 4 | 0 | 9 | 5 | 69.23% | 100% |
| Q23 | 7 | 0 | 6 | 5 | 46.15% | 100% |
| Q24 | 2 | 0 | 11 | 5 | 84.62% | 100% |
| Q25 | 2 | 0 | 11 | 5 | 84.62% | 100% |
| Q26 | 2 | 0 | 11 | 5 | 84.62% | 100% |
| Q27 | 6 | 2 | 7 | 3 | 53.85% | 60% |
| Q28 | 4 | 0 | 9 | 5 | 69.23% | 100% |
| Q29 | 7 | 0 | 6 | 5 | 46.15% | 100% |
| Q30 | 3 | 0 | 10 | 5 | 76.92% | 100% |
| Average | | | | | 71.50% | 94.66% |

*Clarity rate was assessed based on the clarity of the question, simplicity, and the content

[a] number of participants = 13

[b] number of participants = 5

When asked whether they fully understood the questions, all individuals who completed the questionnaires commented on their understanding (Table 2).

In parallel, five experts assess the pre-final version for content validity. The result in (Table 3) showed that the overall rater score yielded CVI 0.68, indicating a problematic rating on the translation in many questions; for example, Q 4, Q 5, Q 14, and Q 27, which need to be restructured and revisited for acceptable cultural relevance [23]. Questions with a CVI above 0.78 were retained unless clarity issues were identified during the interviews, such as Q4, Q5, and Q7. Both overall averages have similar rates and below 80%, and accordingly, several changes in the wording or restructure of some items were made to improve understandability and readability, and the suggestions and combined comments resulted in a final Arabic version.

**Table 3. Evaluation of I-CVIs and agreement for the test and re-test.**

| Raters | Number Of Agreement [a] | | I-CVI [b] | | $P_c$ [c] | | $K$ [d] | |
|---|---|---|---|---|---|---|---|---|
| | n = 5 | n = 3 | n = 5 | n = 3 | n = 5 | n = 3 | n = 5 | n = 3 |
| Question Number | Test | Retest | Test | Retest | Test | Retest | Test | Retest |
| Q1 | 5 | 3 | 1 | 1 | 0.031 | 0.125 | 1 | 1 |
| Q2 | 4 | 2 | 0.8 | 0.67 | 0.156 | 0.375 | 0.76 | 0.47 |
| Q3 | 4 | 2 | 0.8 | 0.67 | 0.156 | 0.375 | 0.76 | 0.47 |
| Q4 | 1 | 2 | 0.2 | 0.67 | 0.156 | 0.375 | 0.05 | 0.47 |
| Q5 | 0 | 2 | 0.2 | 0.67 | 0.031 | 0.125 | -0.03 | 0.47 |
| Q6 | 3 | 3 | 0.6 | 1 | 0.31 | 0.125 | 0.42 | 1 |
| Q7 | 2 | 3 | 0.4 | 1 | 0.312 | 0.125 | 0.13 | 1 |
| Q8 | 3 | 3 | 0.6 | 1 | 0.312 | 0.125 | 0.42 | 1 |
| Q9 | 3 | 3 | 0.6 | 1 | 0.312 | 0.125 | 0.42 | 1 |
| Q10 | 5 | 3 | 1 | 1 | 0.031 | 0.125 | 1 | 1 |
| Q11 | 5 | 3 | 1 | 1 | 0.031 | 0.125 | 1 | 1 |
| Q12 | 4 | 2 | 0.8 | 0.67 | 0.156 | 0.375 | 0.76 | 0.47 |
| Q13 | 5 | 3 | 1 | 1 | 0.031 | 0.125 | 1 | 1 |
| Q14 | 0 | 3 | 0 | 1 | 0.031 | 0.125 | -0.03 | 1 |
| Q15 | 5 | 3 | 1 | 1 | 0.031 | 0.125 | 1 | 1 |
| Q16 | 5 | 3 | 1 | 1 | 0.031 | 0.125 | 1 | 1 |
| Q17 | 5 | 3 | 1 | 1 | 0.031 | 0.125 | 1 | 1 |
| Q18 | 5 | 3 | 1 | 1 | 0.031 | 0.125 | 1 | 1 |
| Q19 | 4 | 3 | 0.8 | 1 | 0.156 | 0.125 | 0.76 | 1 |
| Q20 | 4 | 3 | 0.8 | 1 | 0.156 | 0.125 | 0.76 | 1 |
| Q21 | 3 | 3 | 0.6 | 1 | 0.3125 | 0.125 | 0.42 | 1 |
| Q22 | 4 | 3 | 0.8 | 1 | 0.156 | 0.125 | 0.76 | 1 |
| Q23 | 5 | 3 | 1 | 1 | 0.031 | 0.125 | 1 | 1 |
| Q24 | 3 | 3 | 0.6 | 1 | 0.312 | 0.125 | 0.42 | 1 |
| Q25 | 4 | 3 | 0.8 | 1 | 0.156 | 0.125 | 0.76 | 1 |
| Q26 | 3 | 3 | 0.6 | 1 | 0.312 | 0.125 | 0.42 | 1 |
| Q27 | 1 | 3 | 0.2 | 1 | 0.156 | 0.125 | 0.05 | 1 |
| Q28 | 3 | 2 | 0.6 | 0.67 | 0.312 | 0.375 | 0.42 | 0.47 |
| Q29 | 2 | 2 | 0.4 | 0.67 | 0.312 | 0.375 | 0.13 | 0.47 |
| Q30 | 3 | 3 | 0.6 | 1 | 0.312 | 0.125 | 0.42 | 1 |
| Average of I-CVI | | | 0.68 | 0.92 | | | | |

[a] Number Giving Rating of 3 or 4

[b] I-CVI, item-level content validity index.

[c] $p_c$ (probability of a chance occurrence).

$p_c$ = [N!/A!(N _A)!]* .5N where N = number of experts and A = Number agreeing on good relevance.

[d] $k$* = kappa designating agreement on relevance: $k$ = (I-CVI-pc)/(1-pc).

Some participants interpreted specific terms differently from the actual concept of the questions as a result of the cultural differences between the two languages. The overall clarity was 71%, which means some questions need to be restructured (Table 4) [23]. The outcomes of the participants' remarks during the interview in the first test were summarized in (Table 4) and led to a decision to alter or modify the language phrasing.

Both procedures were reconducted to re-test the final version, which showed an improvement, and similar results indicate that the content and face validity are acceptable. After

**Table 4. Example of interview participant's quotes about questions and the decisions.**

| Item Decision | Questions | Participants' Quote |
|---|---|---|
| Reworded | *Q3* | P1: "I did not understand the word aspect of my care" |
| Restructured | *Q4* | P1: She was not sure about the answer "could you repeat the question?" "I did not get it"<br>P2: "I missed the meaning" |
| Restructured | *Q5* | P3: "I did not understand what the question means?"<br>P4: "I'm not sure. I'm not sure what it is" |
| Reworded | *Q13* | P5: "Not clear about the word aspect; could you explain?" |
| Reworded | *Q20* | P6:" I don't know what this means" |
| Reworded | *Q21* | P7:" what do you mean by working as a team" |
| Restructured | *Q27* | P8: "I understand the opposite of the question; I would think if I was answering this about something completely different" |
| Reworded | *Q29* | P9: "Do you mean by the word staff nurses? "<br>P10: "Cannot interpret the word "struggle," do you mean help between the doctors and nurses?" |

additional review, the last version was presented to three experts, demonstrating acceptable cultural relevance with a CVI of 0.90. The interviews showed that the clarity rate had improved to 94.66%. For any other items that remain with issue, a last review was conducted to finalize the final version.

## Discussion

The present study is the first to report on the psychometrics of the Arabic PMOS-30 in Saudi Arabia using a translated PMOS-30 tool to measure patient perceptions of patient safety. This tool is the only tool that explores the patient's perspective in identifying and improving patient safety. In addition, it was used in a previous study to measure the impact of patient participation in evaluating patient safety in Saudi Arabia: a cross-sectional study [24]. This study sheds light on the role of patients and how their perception is a major contributor to patient safety in general. Patient perceptions play a role in determining patient safety in healthcare. This study is the first to translate and validate the Arabic version of the PMOS-30 questionnaire that was broadly developed to assess patient perceptions. Multiphase tests were conducted to consider the differences in culture and language and to ensure content accuracy, semantic equivalence, and construct validity. Our analysis revealed that the translated tool was acceptable and valid. To adapt a proactive diagnostic tool like PMOS-30, which actively captures the patient's perspective on the safety of care in different languages, we examined several psychometric properties. Our chosen approach for content validity in this study aimed to determine how effectively the translated tool measures the intended content and the extent to which it covers that content. Recognizing the limitations of a single approach, we incorporated multiple methods. This study documents our combined approach, which includes interviewing patients before and after implementing changes and using the Content Validity Index (CVI) with raters both before and after the amendments.

Engaging with patients provided a valuable source of data for potential item rephrasing, resulting in a significant improvement in face validity. The final version of the tool exhibited strong face validity, ensuring the precise interpretation of the results.

In this study, the percentage of the item-CVI was 92%, surpassing the minimum rate of 80% for a newly developed tool [32]. To strengthen our study, the CVI technique was not solely applied; the Kappa evaluation was performed to ensure that the translated items agreed

with the original version and that all the questions were semantically and technically equivalent to the original English questions.

The findings that emerged from the semi-structured interviews were used to assess the questionnaire to minimize the cultural discrepancies between the original and the translated versions. Although the translation did not distort the original meaning, further explanations were required by the researcher during the interviews to clarify the concept. The amendments were derived from the participants' perspectives, who were selected from the proposed target population. The idea of learning from the target population when developing tools has been examined in a related study [33]. Involving service users or a sample of the targeted population revealed potential consequences of including inaccurate results that affect the validity of the outcome. This approach was conducted in other studies, and amendments were conducted among the concerned items according to the service users [33–35]. Therefore, the data obtained from the second round of validation in the qualitative approach is a notable strength of this study to ensure the content validity of the Arabic version. While the literature provides varying recommendations for the number of experts on panels, our study adhered to the guidance of Polit et al. (2007), which suggests that having at least three experts is considered the minimal acceptable number for establishing good evidence of content validity.

One of the limitations of this study is the small sample size examined in both the pilot and the test-retest study. A larger sample may be necessary for further construct validation. This is the first endeavor to assess the PMOS-30 questionnaire in Saudi Arabia, employing randomly selected test subjects with diverse age groups, educational backgrounds, and geographical locations. The sample may not fully represent the entire population of Saudi Arabia. Nevertheless, a diverse sample was selected to diversify the answers and improve the quality of data. This newly translated tool may need minor adjustments to the language if adopted by other Arabic-speaking countries due to variations in Arabic dialects. Furthermore, this Arabic version of the PMOS-30 tool can be used by the majority of Arabic-speaking immigrants in countries like Australia, the United Kingdom, and Europe. Finally, the number of participants is enough to reach data saturation, as data saturation is not necessarily achieved by a larger sample or a smaller sample. The recruitment process was ongoing until the point at which no new data appeared, and data saturation was reached. Data saturation is best achieved by what constitutes the sample size. "For example, one should choose the sample size that has the best opportunity for the researcher to reach data saturation" [35]. Thematic saturation, the point at which new data appear to no longer contribute to the conclusions due to repetition of themes and participant comments, was used to calculate the final sample size [36].

Data generation has stopped at this moment. To obtain data saturation, it has previously been advised that qualitative investigations need a minimum sample size of 12 [26, 37]. Consequently, it was decided that a sample of 13 would be adequate for the qualitative analysis and scope of this study. The sample included only patients receiving inpatient services in different wards and examined their perspectives. Other studies may be needed in the future to examine patients' perspectives on other services, such as long-term care, hospital settings, rehabilitation, and hospice care.

## Conclusions

The PMOS questionnaire translation procedure was conducted in multiple phases and was subjected to I-CVI and the Kappa test. Our findings revealed the importance of multi-structured phases of translation and piloting the outcome.

The expert overcame the suggestions and comments, resulting in the final Arabic version, which was then used to evaluate the content's validity. The final Arabic version of the

questionnaire resulted in the final Arabic version, which was then used to evaluate the content validity. The final Arabic version of the questionnaire appeared to be a reliable tool for measuring patient perceptions of patient safety. The current findings concluded that the PMOS-30 Arabic version questionnaire is a suitable tool to assess patient safety from patient perception in Arabic-speaking hospital settings. Thus, applying validated tools instead of non-validated translations provides results that can be compared with other national and international studies on Patient Safety.

## Supporting information

**S1 Data.**
(CSV)

## Acknowledgments

The authors thank the participants and experts who voluntarily participated in this study.

## Author Contributions

**Conceptualization:** Rabab B. Alkutbe.

**Data curation:** Nawaf M. Alqahtani, Amani K. Al Zahrani, Rawan Alshahri.

**Formal analysis:** Nawaf M. Alqahtani, Rabab B. Alkutbe.

**Investigation:** Amani K. Al Zahrani, Rawan Alshahri, Rabab B. Alkutbe.

**Methodology:** Nawaf M. Alqahtani, Rana Z. Malyani, Rabab B. Alkutbe.

**Software:** Nawaf M. Alqahtani.

**Supervision:** Yasser A. Alaska.

**Visualization:** Amani K. Al Zahrani, Rawan Alshahri.

**Writing – original draft:** Rana Z. Malyani, Rabab B. Alkutbe.

**Writing – review & editing:** Yasser A. Alaska.

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
