## [Decision Letter · Decision Letter 0]

8 Aug 2023

PONE-D-22-31129Evaluating the content and face validity of Arabic-translated Patient Measures of Safety survey PMOS-30PLOS ONE

Dear Dr. Alkutbe,

Thank you for submitting your manuscript to PLOS ONE. After careful consideration, we feel that it has merit but does not fully meet PLOS ONE’s publication criteria as it currently stands. Therefore, we invite you to submit a revised version of the manuscript that addresses the points raised during the review process.

Please read carefully the reviewers feedback and follow their recommendations. Please pay attention especially to the statistical analysis, the sample is very small and the data is not valid without a proper analysis.

We look forward to receiving your revised manuscript.

Kind regards,

Bogdan Nadolu, Ph.D.

Academic Editor

PLOS ONE

Journal Requirements:

Reviewers' comments:

Reviewer's Responses to Questions

**Comments to the Author**

1. Is the manuscript technically sound, and do the data support the conclusions?

Reviewer #1: No

Reviewer #2: Yes

2. Has the statistical analysis been performed appropriately and rigorously? 

Reviewer #1: Yes

Reviewer #2: No

3. Have the authors made all data underlying the findings in their manuscript fully available?

Reviewer #1: No

Reviewer #2: Yes

4. Is the manuscript presented in an intelligible fashion and written in standard English?

Reviewer #1: No

Reviewer #2: Yes

5. Review Comments to the Author

Reviewer #1: 1) All of psychometric characteristics of tool should be examined.

2) The author has mentioned tools for measurement safety patient and satisfaction, it is necessary to mention these tools in the introduction.

3) The authors need to provide a more in-depth discussion about what the constructs in this measure are supposed to be measuring.

4) I would also encourage the authors to again pay close attention to grammar throughout the manuscript

5) The authors need to provide a more in-depth discussion about what the constructs in this measure are supposed to be measuring.

6) The discussion section needs a deeper expansion, it is necessary to compare and discuss all the results with similar studies.

Reviewer #2: Thank you for conducting this interesting study. I have concerns about the followings:

1. There is lack of clarity about content validity in terms of:

• I do not think you have followed Lynns’ Method for content validity? Please follow and refer (Lynn, M.R. (1986) Determination and quantification of content validity. Nursing Research, 35 (6), 382- 385.)

• The underlying context

• The different levels od accepting or deleting each item

• The description of the experts who participated

2. The number of experts is minimum to be accepted

3. The threshold for keeping any item is 0.78 not 0.75

4. Keeping item when it received less than 0.78 I-CVI is not recommended. How have you restructure them to be kept in the questionnaire?

5. The term test-retest is confusing with test-retest reliability so I prefer to use “second round of validation” instead

6. I suggest testing the internal consistency reliability with Cronbach’s alpha test

6. PLOS authors have the option to publish the peer review history of their article (what does this mean?). If published, this will include your full peer review and any attached files.

Reviewer #1: No

Reviewer #2: **Yes: **Yassir A. Yassir

---

## [Author Response · Author response to Decision Letter 0]

12 Nov 2023

Dear All,

Thank you very much for your valuable comments. We really appreciate your effort and time spent on our manuscripts. Please find below the comment and our response to the new line.

Comments Response Line 

Reviewer #1: 

1) All of psychometric characteristics of tool should be examined Thank you very much for your comments. 

Agree it is ideal to test all the psychometric properties it is acceptable in the literature to test partially, like the validity and acceptability.

Also, the psychometric properties were tested in the below reference.

 G, Reynolds C, Moore S, et al. Validation of revised patient measures of safety: PMOS-30 and PMOS-10. BMJ Open 2019;9:e031355. doi:10.1136/ bmjopen-2019-031355 

2) The author has mentioned tools for measurement safety patient and satisfaction, it is necessary to mention these tools in the introduction. Agree, and added. 55- 75

3) The authors need to provide a more in-depth discussion about what the constructs in this measure are supposed to be measuring. Agree, and added. 271-281

4) I would also encourage the authors to again pay close attention to grammar throughout the manuscript. Agree, and added. All over the script 

5) The authors need to provide a more in-depth discussion about what the constructs in this measure are supposed to be measuring Considered in the point 3 

6) The discussion section needs a deeper expansion, it is necessary to compare and discuss all the results with similar studies. Agree, and added. 262, 267, 296-319

Reviewer #2: 

Thank you for conducting this interesting study. I have concerns about the followings:

1. There is lack of clarity about content validity in terms of:

• I do not think you have followed Lynns’ Method for content validity? Please follow and refer (Lynn, M.R. (1986) Determination and quantification of content validity. Nursing Research, 35 (6), 382- 385.)

• The underlying context

• The different levels od accepting or deleting each item

• The description of the experts who participated

 Thank you very much for your valuable comment.

We have not used the Lynn method, however, this tool have been test for it content validity when it was developed. Since this is copyrighted, we get permission to translate and follow the scientific methodology to validate the translation.

We have used 2 methods and we repeated them after the amendments 

1- Patients 

2- Raters 

As we are not permitted to delete items, items who has issues from the second interview were returned to the expert to solve them 

- Added the Expert Description 

144- 148

2. The number of experts is minimum to be accepted Yes however it has been used in the literature 297

3. The threshold for keeping any item is 0.78 not 0.75

 Agree and corrected 

4. Keeping item when it received less than 0.78 I-CVI is not recommended. How have you restructure them to be kept in the questionnaire?

 As we mentioned before, we cannot delete any item for the copyright. We revisit the questions and compare them with patients' results in the table. If there was disagreement we made the changes. 236-237

5. The term test-retest is confusing with test-retest reliability so I prefer to use “second round of validation” instead

 Thank you for your comment, changed accordingly 171, 174, 260

6. I suggest testing the internal consistency reliability with Cronbach’s alpha test It is a good suggestion however, there will be many tests in this paper which could confuse the readers.

---

## [Decision Letter · Decision Letter 1]

8 May 2024

Evaluating the content and face validity of Arabic-translated Patient Measures of Safety survey PMOS-30

PONE-D-22-31129R1

Dear Dr. Alkutbe,

We’re pleased to inform you that your manuscript has been judged scientifically suitable for publication and will be formally accepted for publication once it meets all outstanding technical requirements.

Kind regards,

Bogdan Nadolu, Ph.D.

Academic Editor

PLOS ONE

Additional Editor Comments (optional):

Reviewers' comments:

Reviewer's Responses to Questions

**Comments to the Author**

1. If the authors have adequately addressed your comments raised in a previous round of review and you feel that this manuscript is now acceptable for publication, you may indicate that here to bypass the “Comments to the Author” section, enter your conflict of interest statement in the “Confidential to Editor” section, and submit your "Accept" recommendation.

Reviewer #3: All comments have been addressed

2. Is the manuscript technically sound, and do the data support the conclusions?

Reviewer #3: Partly

3. Has the statistical analysis been performed appropriately and rigorously? 

Reviewer #3: Yes

4. Have the authors made all data underlying the findings in their manuscript fully available?

Reviewer #3: Yes

5. Is the manuscript presented in an intelligible fashion and written in standard English?

Reviewer #3: Yes

6. Review Comments to the Author

Reviewer #3: The comments have been responded by authors satisfactorily. I recommend the authors to attach the related checklist for valididty study according to equatornetwork.com.

7. PLOS authors have the option to publish the peer review history of their article (what does this mean?). If published, this will include your full peer review and any attached files.

Reviewer #3: No

---

## [Editor Report · Acceptance letter]

22 May 2024

PONE-D-22-31129R1 

PLOS ONE

Dear Dr. Alkutbe, 

I'm pleased to inform you that your manuscript has been deemed suitable for publication in PLOS ONE. Congratulations! Your manuscript is now being handed over to our production team.

Kind regards, 

on behalf of

Dr. Bogdan Nadolu 

Academic Editor

PLOS ONE